# Trans-marine dispersal inferred from the saltwater tolerance of lizards from Taiwan

**Min-Hao Hsu**[1,2], **Jhan-Wei Lin**[1], **Chen-Pan Liao**[1,3], **Jung-Ya Hsu**[1], **Wen-San Huang**[1,2,3]*

**1** Department of Biology, National Museum of Natural Science, Taichung, Taiwan, **2** Department of Life Sciences, National Chung Hsing University, Taichung, Taiwan, **3** Department of Life Science, Tunghai University, Taichung, Taiwan

* wshuang.380@gmail.com

## Abstract

Dehydration and hypersalinity challenge non-marine organisms crossing the ocean. The rate of water loss and saltwater tolerance thus determine the ability to disperse over sea and further influence species distribution. Surprisingly, this association between physiology and ecology is rarely investigated in terrestrial vertebrates. Here we conducted immersion experiments to individuals and eggs of six lizard species differently distributed across Taiwan and the adjacent islands to understand if the physiological responses reflect the geographical distribution. We found that *Plestiodon elegans* had the highest rate of water loss and the lowest saltwater tolerance, whereas *Eutropis longicaudata* and *E. multifasciata* showed the lowest rate of water loss and the highest saltwater tolerance. *Diploderma swinhonis*, *Hemidactylus frenatus*, and *Anolis sagrei* had medium measurements. For the eggs, only the rigid-shelled eggs of *H. frenatus* were incubated successfully after treatments. While, the parchment-shelled eggs of *E. longicaudata* and *D. swinhonis* lost or gained water dramatically in the immersions without any successful incubation. Combined with the historical geology of the islands and the origin areas of each species, the inferences of the results largely explain the current distribution of these lizards across Taiwan and the adjacent islands, pioneerly showing the association between physiological capability and species distribution.

## Introduction

Over-water dispersal makes terrestrial organisms move from one land mass to another. The capability and probability of this dispersal ecologically relate to the distribution of the creatures across lands and islands and then further evolutionarily influence biogeography and biodiversity [1–3]. Molecular and fossil evidence indicates that organisms may only spread outward between lands by crossing the ocean, rather than using land bridges [2,4,5], denoting that over-water dispersal is an important strategy for organisms. In the Anthropocene, although human transportation has become a path to crossing oceans for many species now recorded as introduced species, for a species invading new areas artificially, they could still spread out to adjacent areas naturally [6]. For example, Norway rats, which invaded islands around the world by boats, dispersed to adjacent islands by natural drifting or swimming [7,8]. Therefore,

---

**Data Availability Statement:** All relevant data are within the manuscript and its Supporting Information files.

**Funding:** Ministry of Science and Technology, Taiwan MOST 109-2621-B-178-001-MY3 to

corresponding author Wen-san Huang. he funders had no role in study design, data collection and analysis, decision to publish, or preparation of the manuscript.

**Competing interests:** The authors have declared that no competing interests exist.

over-water dispersal is not only important for understanding the distribution of native species but also plays a substantial role in determining the range an invasive species can reach.

Extreme saline environments and crucial water loss are major physical challenges for terrestrial organisms when they are drifting on the ocean [9–11]. High-salinity environments are a stress for organisms. Different species have their own salinity tolerances [11–13]; this variation influences the survival differences among species in marine floating, leading to the variance in ability of oversea dispersal at a species level that further influences the distribution of the species [13,14]. However, research about the association between this physiological ability and distribution has mainly focused on plants, freshwater animals, and arthropods [e.g. 10,15,16]. The dispersal of terrestrial vertebrates is rarely explored. In reptiles, some species appear in saltwater wetlands or coasts, even surviving there for a long time [17–19], demonstrating that they can tolerate higher salinity environments. Nevertheless, previous studies about salinity tolerance have mostly focused on marine or estuary species [18,20,21]. In the few studies on freshwater and terrestrial species, the salinity treatments were usually orally fed or injective [12,22], whereas treatment with saltwater contact, the most relevant situation faced by a drifting individual, has rarely been conducted in previous literature. Furthermore, none of these studies link salinity tolerance to species distributions. Consequently, this key ability for crossing oceans, the saltwater tolerance of direct water contact and how it relates to distribution, has not been fully elucidated.

Dehydration is the most crucial challenge for organisms drifting on the ocean and the vital factor of saltwater tolerance. In reptiles, water loss occurs primarily through skin in addition to the loss through respiration [23,24]. The water loss of the skin is related to the adaption to the climate of the habitat environment [25–27], whereby species living in hot and dry habitats lose water slower than those living in cold and wet habitat [27,28]. In addition, the lower rate of water loss indicates that the species can retain water for a long time, which could delay death from dehydration [29,30]. Thus, species with lower rates of water loss are more likely to survive a drifting period of over-water dispersal than species that lose water fast. However, the relationship between water loss and survival rate in the context of ocean crossing in vertebrates has rarely been reported.

For most reptiles, there are two different ontogenetic stages to ocean crossing: the individual stage and the egg stage. Currently, ocean crossing at the individual stage has been reported abundantly through swimming, floating, or rafting [1,31–33]. Compared with large species which can swim for a long distance, floating and rafting are the more likely ways for small reptiles to cross oceans [1,34]. In geckos, dispersal by natural rafts or boats has been suggested in some small species [35,36]. Regarding *Anolis*, it can float for a short time on sea water [34] to disperse between islands in their native places [1,4]. On the other hand, it is probable that over-water dispersal occurs at the egg stage [35,37]. The saltwater tolerance and water loss of reptile eggs may be majorly determined by the type of eggshell. Reptilian eggs are mainly divided into two types: parchment-shelled eggs and rigid-shelled eggs [38]. The shell of the former is thinner and less calcareous than that of the latter, resulting in differences in flexibility and permeability of water exchange. Parchment-shelled eggs are highly sensitive to environmental humidity [39], whereas rigid-shelled eggs, which have a dense and hard shell, could limit the exchange of water and substance from the environment [40]. Therefore, rigid-shelled eggs should have higher tolerance to seawater. Some gecko studies have shown that eggs can tolerate sea water well after immersion treatment [41,42]. For parchment-shelled eggs, successful incubation after immersion of sea water has only been reported in *Anolis sagrei* [43].

Taiwan and the adjacent islands are separated from the Asian continent by a sea strait with a depth of approximately 70 meters. However, the main island of Taiwan and the small islands between Taiwan and China were all connected to the Asian mainland before 1.55 Ma and

during the Last Glacial Maximum in 26.5–18 ka, during which the sea level fell 135 meters (S1 Fig A in S1 File) [44–46]. Therefore, Taiwan and these western islands are continental islands. Terrestrial species could disperse between Taiwan and Asian mainland through the land bridge (S1 Fig A in S1 File). In contrast, the eastern ocean of Taiwan is drastically deep (a depth of over 1,000 meters) due to the oceanic trench in the nearby eastern coast, preventing the islands east of Taiwan from connecting to any island and mainland historically (S1 Fig A in S1 File). Thus, the eastern islands are oceanic islands, where the species arrive only by over-water dispersal. Combining these historical contexts of geographic connection across Taiwan and these islands, knowledge of the variation of salinity tolerance among species could demonstrate how the current species distributions were formed.

In this study, we aimed to examine the heterogeneity of saltwater tolerance of small lizard species across Taiwan and the adjacent islands to determine their potential ability of over-water dispersal and then inspect the current distributions of these species and their saltwater tolerance within the context of historical geology. Six small lizard species, specifically four native species and two introduced species, were chosen in this study (Table 1 and S1 Fig B-G in S1 File). We performed half-immersion experiments on these six species and their eggs to simulate the extreme rafting condition, directly contact with the water. Specifically, we first examined the water loss and the survival rate/incubation rate of these small lizards and their eggs using the half-immersion experiments to evaluate the possibility of natural cross-ocean dispersal. Second, we inspected the association between these variations in saltwater tolerance and the distributions of these six species with the historical contexts of geographic connection across Taiwan and the adjacent islands. Finally, in addition, we assessed the dispersal risk of the two introduced species for conservation purposes.

## Materials and methods

### Sample collection and husbandry

Six species, *P. elegans*, *E. longicaudata*, *E. multifasciata*, *D. swinhonis*, *H. frenatus*, and *A. sagrei*, were collected and kept individually in a plastic tank (36 cm × 17 cm × 20 cm) with water, food, and a natural light-dark cycle in a laboratory. *P. elegans* and *H. frenatus* were collected from the Pakua Mountain Range (23.989–24.024˚N, 120.576–120.596˚E) in Changhua. *D. swinhonis* was collected from Dadu Mountain (24.156˚N, 120.559˚E) in Taichung. *E. longicaudata* and *E.*

**Table 1. Distributions of six lizard species in Taiwan and adjacent islands.**

| | Taiwan | Penghu | Little Liuqiu Island | Guishan Island | Green Island | Orchid Island | Philippines |
|---|---|---|---|---|---|---|---|
| Distance from Taiwan and estimated duration of drifting/rafting | NA | 47 km, west (26.1 hr) | 13 km, southwest (7.2 hr) | 11 km, northeast (6.1 hr) | 44 km, southeast (24.4 hr) | 63 km, southeast (35 hr) | 161 km, southeast (89.4 hr) |
| *Plestiodon elegans* | O | O | O | O | X | X | X |
| *Eutropis longicaudata* | O | X | O | X | O | O | X |
| *Eutropis multifasciata* | O | X | O | X | O | O | O |
| *Diploderma swinhonis* | O | X | O | X | O | O | X |
| *Hemidactylus frenatus* | O | O | O | O | O | O | O |
| *Anolis sagrei* | O | X | X | X | X | X | X |

The distance is the closest distance to Taiwan, and estimated duration of drifting/rafting by sea current of lizards. The direction is the relative position to Taiwan. The character 'O' indicates that this species exists in that location, and the character 'X' indicates no distribution. The estimated duration of drifting/rafting was estimated according the mean speed 0.5 m/s of the current pass each island [Penghu: 0.1–1 m/s [47]; Little Liuqiu Island: 0.3–0.8 m/s [48]; Guishan Island, Green Island, Orchid Island, and Philippines: 0.4–0.6 m/s [49]]. The asterisk marked indicates that the species was introduced species.

*multifasciata* were collected from Sandimen (22.736–22.724˚N, 120.638˚E) and Nanhe (22.443–22.453˚N, 120.621–120.623˚E) in Pingtung. *A. sagrei* was captured from Qixingtan (24.019˚N, 121.626˚E) in Hualien. All captured individuals were kept in standard conditions for one week for acclimation before performing the lizard experiment. The pregnant *E. longicaudata* and *D. swinhonis* females were kept separately and checked frequently for eggs, which were collected while fresh. The *H. frenatus* eggs were collected from the wild because the pregnant females failed to lay eggs in the laboratory. These eggs were used in the egg experiment. After the experiments, all of the lizards were released back to the places of capture. For the principle of reduction in animal experiment, we minimized the number of experimental individuals to 6–10 to evaluate saltwater tolerance. These data could increase our understanding of their cross-water potential which would be helpful in future conservation management for these species, especially the two introduced species. During the experiment, any individual observed having oral secretions in the mouth and the nose and the activity declined severely, we stopped the experiment and replaced it back to standard condition of husbandry (in a plastic tank at normal temperature with water, food, and a natural light-dark cycle in a laboratory) to recover and evaluate if euthanasia is needed. The individuals which did not recover within 24 hours were euthanized with carbon dioxide. The recovered individuals would not be used again and were released back to wild after the experiment (except the two invasive species). The inspection of animal health and behavior was conducted two times per day during the experiment. All of the collection, husbandry and subsequent treatment procedures followed the Wildlife Conservation Act and Animal Protection Act of Taiwan. The animal use protocols were strictly reviewed and approved by IACUC in the National Museum of Natural Science (license No. 1061701832).

## Half-immersion experiment

We used half-immersion in all experimental treatments to mimic the most extreme rafting situation which individual directly contacts with seawater but not fully floating on the water. In the lizard experiment, individuals of each species were randomly separated into three groups, which received the saltwater (SW), freshwater (FW), or control treatments. Individuals with SW were placed in a 36 cm × 17 cm × 20 cm tank with 3.5% SW. The water surface slightly exceeded the abdomen of the lizard but did not exceed the mouth by controlling the water level in the tank to have it shallow enough for the animal's feet to touch the bottom. In FW, individuals were treated as the setting in SW but replaced SW with FW. For the control treatment, individuals were placed into a tank without water. Each individual was weighed before the treatment and daily in the three continuous days of treatment. During the experiment, the faeces and shedding scales of each individual were collected and weighed as well. For the species with toepads, *H. frenatus* and *A. sagrei*, We used Vaseline on the wall of the tank to prevent the lizard from sticking to the wall. The temperature of this experiment was controlled at approximately 27˚C, and the humidity was controlled between 45–55%. In total, 91 *P. elegans*, 49 *E. longicaudata*, 62 *D. swinhonis*, 61 *H. frenatus*, 93 *E. multifasciata*, and 60 *A. sagrei* individuals were used in this experiment. There were at least ten individuals of each species in each treatment, except for male *E. longicaudata* (either 6 or 7) because they were difficult to collect.

In the egg experiment, only eggs of *E. longicaudata*, *D. swinhonis*, and *H. frenatus* were used. Eggs which had been through at least half of their respective incubation days were used because the developmental time of the reptile eggs would influence the tolerance to environmental change [50]. The incubation periods were approximately 35 d in *E. longicaudata* and 50 d in *D. swinhonis*. The eggs of these two species were laid and collected in the laboratory and then kept in 27˚C and 45–55% RH for 17 and 25 d, respectively, before the egg experiment. The incubation period in *H. frenatus* was 56 d, but most of the individuals were collected from the wild without

a specific laying date; they were used in the egg experiment one week later from the date of collection. However, the incubation days in all samples were from 14 to 35 d (24.7 ± 9.1 d) after the experiments; this range is close to 28 d, midpoint of the incubation period. Eggs of each species were randomly separated into three groups as the design of the lizard experiment. After measuring the lengths and widths, the eggs were half buried in the wet culture soil in the control treatment. In the FW and SW treatment, eggs were half immersed directly. Each egg was weighed before the treatment and daily in the three continuous days of treatment. This process took three days. In total, 32 eggs of *E. longicaudata*, 30 eggs of *D. swinhonis*, and 9 eggs of *H. frenatus* were used in the egg experiment. There were at least ten individuals of each species in each treatment, except *H. frenatus* (only three) because they were difficult to collect.

We calculated the water loss and the survival rate of the individuals and eggs to determine the effect of treatment. We defined the day that the experiment started as the initial day and the last treatment day as the third day. After the experiments, we calculated the rate of total water loss (RTWL), $(D_0 - D_3 - D_w) / D_0$, where $D_0$ is the weight of the initial day, $D_3$ is the weight of the third day during the experiment, and the $D_w$ is the wastes including faeces and shedding skin, which was not used in the egg experiment.

## Statistical analysis

We performed Bayesian linear models and Bayesian logistic models to separately fit the RTWLs and survival/hatching rates. For fitting the RTWLs/survival rates of lizard, we considered the following three factors/covariates: 1) a merging factor which merged species, sex and salinity treatments, 2) scaled mass indexes (SMIs) and 3) the interaction between the two main factors/covariates above. The SMI term was an indicator of body condition accounting for the SVL and weight of lizards and calculated according to Peig & Green [51]. Notably, we excluded the data of the SW group and those of all lizards died before day three when fitting the RTWLs, because most of the lizards did not survive before the end of the SW experiments. For fitting the RTWLs and hatching rates of eggs, we considered the following three factors/covariates: 1) a merging factor which merged egg species and salinity treatments, 2) the weights of eggs and 3) the interaction between the two main factors/covariates above. Both the interaction between SMI and the sex-stage-treatment factor and the effect of SMI were highly unsupported in the RTWL and survival rate of the lizards (RTWL: Bayes factor [BF] < 0.01 in the interaction, BF = 0.02 in SMI; survival rate: BF < 0.01 in the interaction, BF = 0.25 in SMI); similarly, both the interaction between weight and sex-stage-treatment factor and the effect of weight were highly unsupported in the RTWL and hatching rate of the eggs (RTWL:BF < 0.01 in the interaction, BF = 0.17 in weight; hatching rate: BF = 0.68 in the interaction, BF = 0.52 in weight); thus, we eliminated these factors in the following results. We performed equivalence tests to compare the survival/hatching rates and RTWLs among sex-stages/treatments/species. We defined the region of practical equivalence (ROPE) as ±0.2 Cohan's *d* for comparing RTWLs and as1 ± 1.2 odds ratio for comparing hatching/survival rates. All Bayesian models were performed via the R package 'brms' [52] under the R framework (version 3.3.1). For each model, ten parallel Markov chain Monte Carlo (MCMC) chains were performed, and 1,000 MCMC samples were collected after 5,000 burn-in iterations per chain. We assigned weak priors to sufficiently cover the potential range of the parameters. The convergences of the MCMC samples were confirmed visually.

## Results

### Rate of water loss of lizards

Among the six species, the *P. elegans* individuals had the highest rate of total water loss (RTWL) in the control (Fig 1A), indicating that this species loses water fast in the air. In *E.*

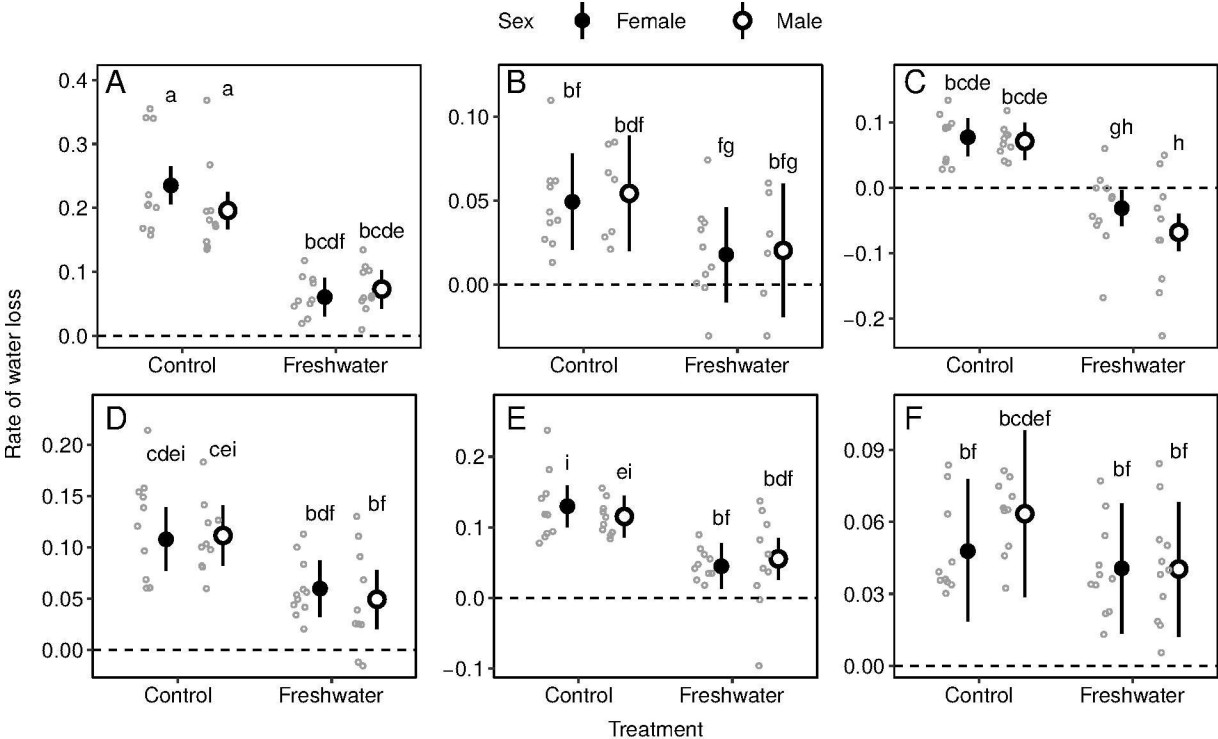

**Fig 1.** Rates of total water loss (RTWLs) in six lizard species, *P. elegans* (A), *E. longicaudata* (B), *D. swinhonis* (C), *H. frenatus* (D), *A. sagrei* (E), and *E. multifasciata* (F) in Taiwan. Circles and vertical bars present the posterior mean and 95% credible intervals on RTWLs, respectively. Alphabet letters above bars presents the results of post hoc comparisons; two groups sharing same letter(s) indicate no significant difference.

*longicaudata* and *E. multifasciata*, the RTWLs of the control were low, and there were no significant differences between the control and freshwater treatments (FW) (Fig 1B and 1F). Meanwhile, in the other four species, RTWLs of FW were significantly lower than that of control (Fig 1A, 1C, 1D and 1E). In FW, except for the negative value of *D. swinhonis* (male: −0.069 ± 0.087, female: −0.032 ± 0.058; Fig 1C), which indicated that *D. swinhonis* drank water, the RTWLs of the other five species were similar (male: 0.021–0.073, female: 0.019–0.063; Fig 1A, 1B, 1D, 1E and 1F). The RTWLs were not different between males and females in all treatments in all species. Because most of the individuals in the saltwater treatments (SW) died before the end of the treatment period, resulting in the scarcity of RTWL data in this group, we did not include them in the comparison in this study. The RTWLs of the two suervived species in SW treatment were listed here. *E. longicaudata*: male: 0.040 ± 0.049; female: 0.036 ± 0.032. *E. multifasciata*: male: 0.082 ± 0.066, female: 0.037 ± 0.078)

## Survival rate of lizards

The survival rates (90–100%) to the end of the treatment period were not different between the control and FW in all species (Fig 2). In contrast, the survival rates of SW decreased significantly compared with the control and FW in most of the species, except *E. longicaudata* and *E. multifasciata* (Fig 2). When treated with SW, the survival rates of the four influenced species were extremely low, indicating their low tolerance to saltwater (Fig 2A, 2C, 2D and 2E). Meanwhile, the survival rates of the two *Eutropis* species were not significantly affected by saltwater immersion (Fig 2B and 2F). Sex was not associated with the survival rate in all treatments in all species.

Table 2 shows the survival in detail. *P. elegans* survived worse even during the treatment period where 90% of individuals in SW died on day one (Table 2). In contrast, the survival

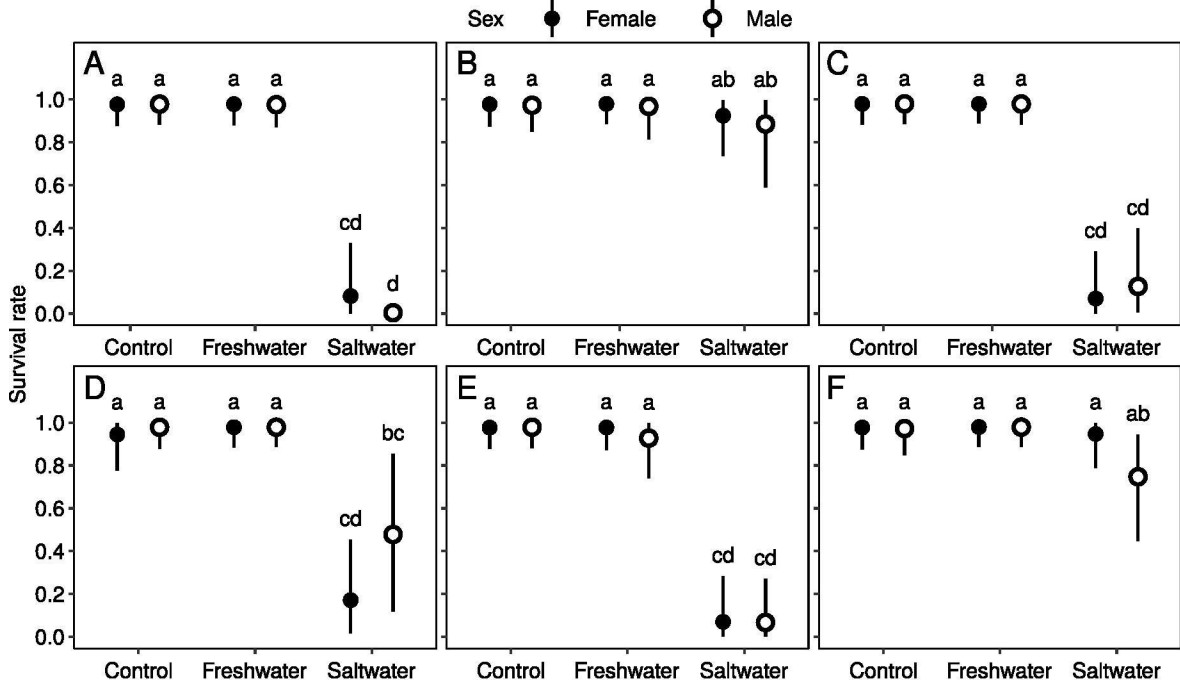

**Fig 2.** Survival rates in three sequent days in six lizard species, *P. elegans* (A), *E. longicaudata* (B), *D. swinhonis* (C), *H. frenatus* (D), *A. sagrei* (E) and *E. multifasciata* (F), in Taiwan. Circles and vertical bars present the posterior mean and 95% credible intervals on the survival rate, respectively. Letters above the bars present the results of the post hoc comparisons; two groups sharing the same letter(s) indicate no significant difference.

rates of *E. longicaudata* and *E. multifasciata* remained higher than 70% at the end of the treatment. In *D. swinhonis*, more than 70% of SW individuals survived on the first day; however, most of them died on the second day, and only 1 out of 20 SW individuals survived to the end of the treatment. In *H. frenatus*, the survival rate of SW was higher than 60% on the first day and 40% on the second day, and 4 out of 20 individuals survived to the end. The SW survival rate of *A. sagrei* was higher than 70% on the first day, and it was also 40% on the second day. None of them survived to the end of the SW treatment.

**Table 2. Accumulated survival rates in six lizard species in the saltwater treatment.**

| Species | Sex-stage | N | Survival rate (%) | | |
|---|---|---|---|---|---|
| | | | Day1 | Day2 | Day3 |
| *P. elegans* | male | 10 | 0 | 0 | 0 |
| | female | 10 | 10 | 0 | 0 |
| *E. longicaudata* | male | 6 | 100 | 83.3 | 83.3 |
| | female | 10 | 100 | 100 | 90 |
| *D. swinhonis* | male | 10 | 90 | 10 | 10 |
| | female | 10 | 70 | 0 | 0 |
| *H. frenatus* | male | 10 | 60 | 40 | 30 |
| | female | 10 | 80 | 40 | 10 |
| *E. multifasciata* | male | 10 | 100 | 90 | 70 |
| | female | 11 | 100 | 100 | 90.9 |
| *A. sagrei* | male | 10 | 90 | 60 | 0 |
| | female | 10 | 70 | 40 | 0 |

### Rate of water loss of eggs

In the eggs of *E. longicaudata* and *D. swinhonis*, the RTWLs in the control and FW were lower than zero (Fig 3A and 3B), indicating that the eggs in these two treatments absorbed water during the experimental treatments. The significantly lower RTWL in FW denoted that they took in much more water than the control. Meanwhile, higher RTWLs in SW than in the control and FW were shown in both species that lay parchment-shelled eggs. (Fig 3A and 3B), indicating that the eggs of these two species lost water severely during SW. In contrast, the RTWLs in *H. frenatus* had no significant difference among treatments and were not different from zero, indicating that none of the treatments influenced the water within the eggs, and almost no net water loss/absorption occurred in these rigid-shelled eggs (Fig 3C).

### Hatching rate of eggs

In *E. longicaudata* and *D. swinhonis*, the hatching rates were similar between the control and FW (Fig 3D and 3E), but those in SW were significantly lower, indicating that saltwater immersion indeed decreased the hatching success rates for these two species. In contrast, in *H. frenatus* the hatching rate in the control treatment (100%) was not different from that in FW (100%) and SW (100%; Fig 3F), denoting that none of the treatments affected the hatching success rates of these rigid-shelled egg.

### Discussion

Our results clearly show the heterogeneity of saltwater tolerance among lizard species in both the individual and egg stage that may further reflect the ability of over-ocean dispersal at

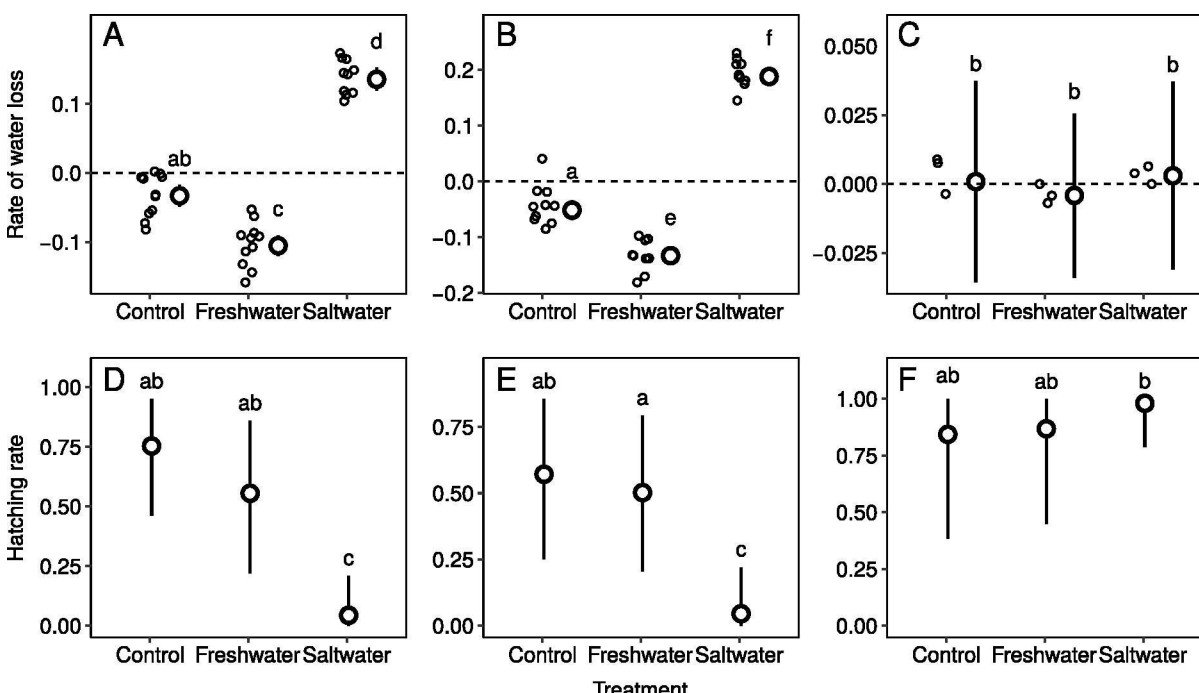

**Fig 3.** Rates of water loss (RTWLs, A-C) and hatching rates (D-F) of eggs in three lizard species in Taiwan. RTWLs of eggs of *E. longicaudata* (A), *D. swinhonis* (B), and *H. frenatus* (C). The hatching rates of eggs of *E. longicaudata* (D), *D. swinhonis* (E), and *H. frenatus* (F) in three days. Circles and vertical bars present the posterior mean and 95% credible intervals of RTWLs, respectively. Letters above the bars present the results of the post hoc comparisons; two groups sharing the same letter(s) indicate no significant difference. Post hoc comparisons of RTWLs and hatchling rates were analysed separately.

interspecies level. Combined with the historical geology and the place of species origin, our results mainly interpret the distributions of these species across the islands around Taiwan and how they arrive at Taiwan (Fig 4). This is the first report testing the saltwater tolerance of terrestrial vertebrates with half-immersion experiments and possible connections of their physiological traits to their dispersal abilities.

## Saltwater tolerance and water loss

Among the native species, *P. elegans* is less likely to cross oceans, whereas *E. longicaudata* tolerated SW notably well, indicating that this species is more able to spread by drifting on the ocean. The other two native species, *D. swinhonis* and *H. frenatus*, had moderate tolerance to SW. Compared with *P. elegans*, they may have better ability to cross oceans if the floating period is shorter than three days. In introduced species, *E. multifasciata* had potential to disperse across oceans as the native *Eutropis* in Taiwan. In contrast, *A. sagrei* tolerated SW the worst, therefore they could not drift on the ocean for a long time as the native *D. swinhonis* and *H. frenatus*. If they could cross the ocean to land within two days by drifting, for example crossing small straits, island series, and archipelagos, over-water dispersal is still possible. That may be why *A. sagrei* has been observed entering the sea and arrive to adjacent islands [1].

The variance in SW survival rate among species may be determined by the difference in RTWL. Although we did not have these data in SW because of the high mortality, the RTWL values in the control and the differences between the control and FW still provides a glimpse that the amount of water loss varied among species and likely occurs in SW. Mechanically, this variance in water loss may be attributed to the differences in species body size [56–58], the amount of lipid content under the skin [25,26,57], and the morphology of scales [59]. We do not know the lipid contents among these species, but we did find that the size of the scales is relatively larger in *E. longicaudata* and *E. multifasciata* than in other species examined in this study(data not reported). Meanwhile, body size is not the likely factor in this study because it was not significant in the RTWL analysis. Evolutionarily, phylogeny [23,56] and adaptation to the habitat climate (slower water loss in species living in hotter and drier habitats) [27,28] may be associated with the variance in RTWL as well. *Plestiodon* originated in temperate northeast Asian [53]; thus, it is physiologically adapted to low temperatures, whereas the other species originated in tropical areas [54,60] with high temperatures. This long-term effect might lead to higher rates of water loss in *P. elegans* than in the other species.

In the egg experiment, the eggs of *E. longicaudata* and *D. swinhonis* tolerated SW worst. In contrast, the low RTWL and the high hatchling rate in *H. frenatus* regardless of the treatment indicate that the eggs of this species tolerate SW well. Gecko eggs cope with seawater well in both the experiment without gas exchange (full immersion) [61] and that with air contact (eggs surrounded by a wet gauze) [41] in previous studies, suggesting that these eggs may be likely to survive in both floating and rafting on the ocean. The difference in SW tolerance of the eggs among these three species may have resulted from the types of egg shell; the eggs of *E. longicaudata* and *D. swinhonis* are parchment-shelled, whereas those of *H. frenatus* have a rigid shell. The RTWL results in this study are consistent with previous studies showing that water permeability is usually higher in the former than in the latter [40]. This variance may largely determine the egg tolerance to SW and the heterogeneity of ocean-crossing ability.

## Distribution and dispersal routes

Together with the historical topography around Taiwan, this study largely reflects the current distribution of each species (S1 Fig B-G in S1 File) and could infer the dispersal history of some species (Fig 4). The low saltwater tolerance of *P. elegans* in this study explains why this

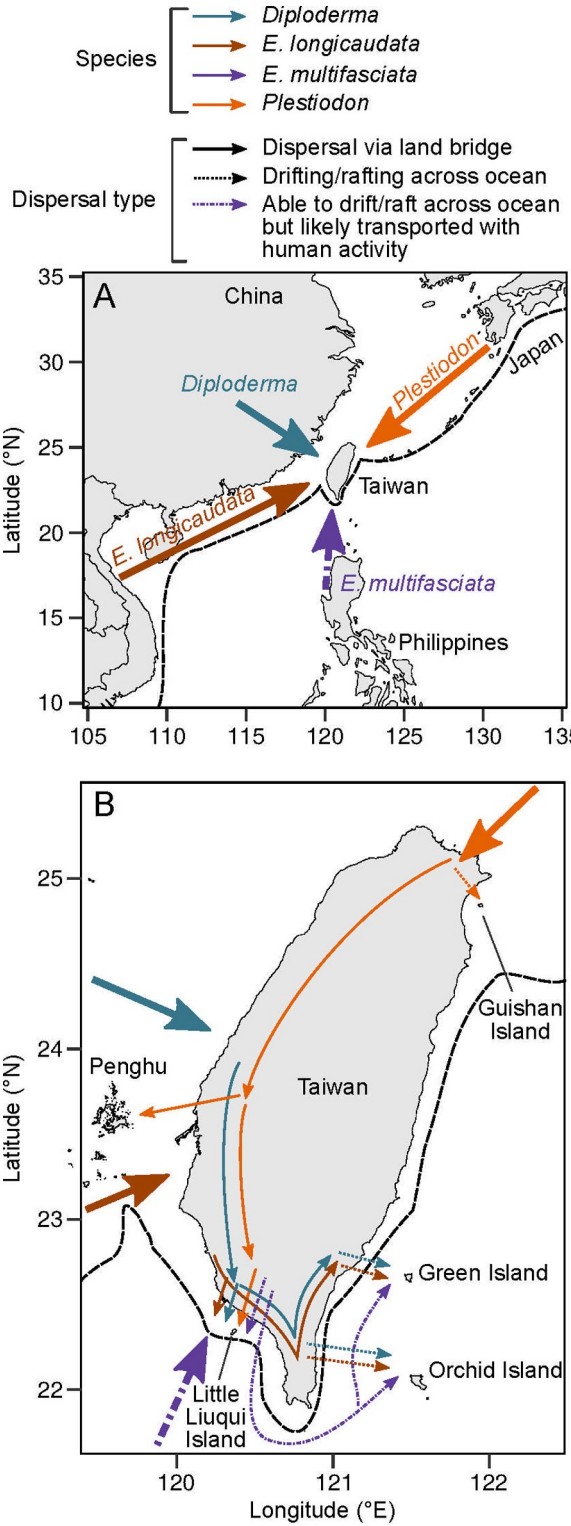

**Fig 4. Dispersal routes of lizard four species in Taiwan.** A. Dispersal routes to Taiwan. B. Dispersal route between Taiwan and the surrounding islands. The orange arrow represents the routes of *Plestiodon* [53]. The brown represents *E. longicaudata* [54]. The blue green represents *Diploderma* [55]. The purple dashed arrow represents the possible route of *E. multifasciata*. The solid arrows represent the possible dispersal routes on land. The dotted arrows represent the possible drifting/rafting routes across ocean. The black dashed line represents the coastline before 1.55 Ma. The coastlines were based on the open data from Natural Earth and Taiwan's Ministry of the Interior, which were licensed

species is only distributed on the western islands, Penghu and Little Liuqiu Island, but not on Green Island and Orchid Island (S1 Fig B in S1 File); arrival to the oceanic islands must be through over-water dispersal. It may further explain why *Plestiodon* species are almost exclusively distributed on continent and continental islands [53,62]. *P. elegans* diverged with a related species that existed in Japan and the Ryukyu Islands approximately 4.23 Ma [53]. At that time, the Ryukyu Islands and Taiwan were still connected to the East Asian continent. The ancestral species may have spread southward from the Ryukyu Islands to Taiwan via the land (Fig 4A) and then spread to the western islands without crossing the ocean (Fig 4B).

The high saltwater tolerance of *E. longicaudata* demonstrates that this species has the ability to raft on the ocean for long distances, explaining why this species distributes in the oceanic islands nearby Taiwan (S1 Fig C in S1 File). This species evolved in mainland Southeast Asia approximately 38 Ma [54], possibly arriving in Taiwan via the land bridge when Taiwan was still connected to the Asian continent (Fig 4A), then dispersed to the ocean islands by oceanic driftings or natural rafts (Fig 4B). Currently, *E. longicaudata* does not occur in northern Taiwan because of the limitation of low winter temperatures. However, if climate warming continues, it may spread northward in Taiwan and possibly cross the ocean to the north islands near Taiwan.

*D. swinhonis* may survive on the ocean for only one day according to our results, while they have a wide distribution, including two oceanic islands, Green Island and Orchid Island (S1 Fig D in S1 File). This genus originated on mainland China and may dispersed via two routes: dispersal through the land bridge and over-water dispersal to the islands where they could arrive within one day (Fig 4A and 4B). Because the distances between Taiwan and the adjacent islands are relatively short (Table 1), force from climate events, such as the monsoon or typhoons, may shorten the dispersal duration to these islands to single day, even the farthest Orchid Island. Therefore, their distributions on the western island could be explained by both land bridge dispersal and over-water dispersal, whereas the eastern island distributions resulted exclusively from the route across the ocean.

A noteworthy conflict between the results and current distribution occurs at nothern Guishan Island. Guishan Island emerged above the ocean approximately 7,000 years ago [63] and was never connected to Taiwan by a land bridge. *P. elegans*, which has very low tolerance to saltwater, occurs on this island (S1 Fig B in S1 File), while *D. swinhonis*, which has moderate saltwater tolerance, does not (S1 Fig D in S1 File). In addition, *Diploderma* species dispersed across the Ryukyu Islands through the Kuroshio current [5] but not on Guishan Island, which is also on the route of Kuroshio. One explanation is that over-water dispersal is a rare event for both species while it only accidentally occurred in *P. elegans*. Alternatively, perhaps the microhabitat on this small island is suitable for *P. elegans* but not for *D. swinhonis*; the former appears in dense bush, but the later needs forest. There is a similar situation in Penghu, which is located in the land bridge (S1 Fig A in S1 File). *P. elegans* exists there, but *D. swinhonis*, which originated from China, does not [55] (S1 Fig B and D in S1 File). The environment of Penghu may not be suitable for *D. swinhonis* to survive permanently because the trees are too few for this arboreal species.

*H. frenatus* has the widest distribution across islands, which is not in line with its moderate saltwater tolerance, but it is in accordance with the excellent egg tolerance to sea water in this study. Now it is hard to know where this species originated because they frequently disperse through human activity [36,64,65]. In this study, we showed the high potential to disperse via

ocean crossing during the egg stage, matching the speculation of a previous study [35,37]. Though we could not exclude the possibility of human translocation, the high SW tolerance of the gecko eggs shown in this study might partially explain the wide distribution and low genetic divergence of the populations of this species across lands and islands in Southeast Asia [60].

In addition, we also showed that the two invasive lizard species in conservation issues; *E. multifasciata* has the potential to disperse to any islands naturally, and it should be carefully monitored in the future. For *A. sagrei*, we should focus on the islands near Taiwan to prevent natural over-water dispersal and the islands that have flourishing agricultural transportation. *E. multifasciata* is thought to be introduced to Kaohsiung in southern Taiwan by artificial transportation since 1994 [66], and now they have spread outward to central Taiwan and some adjacent islands (S1G Fig in S1 File). The high saltwater tolerance of *E. multifasciata* in our results showed that they are able to disperse across ocean to these islands from Taiwan, or even from Philippines by the Kuroshio current, because it could survive (>70%) lasting for 3 days (Table 2 and Fig 2), and some individuals even survived after 5 days in half-immersion treatment (unpublished data) which over the duration in rafting needed from Phillipines to Taiwan (89.4 hrs; Table 1). However, we still suggest that they are introduced by human activity (Fig 4A and 4B) because they are found on these islands recently (2008 on Green Island; 2017 on Orchid Island) and densely live around area nearby the harbour and airport. More evidence is needed for confirming how the species arrived on Taiwan and these islands. The point is that colonizing other islands near Taiwan is highly possible for them in the future, even the northern islands if the climate warming continues. On the other hand, *A. sagrei* are believed to be introduced to Taiwan by human transportation, too [67]. The medium saltwater tolerance of *A. sagrei* in our results showed that it may drift and survive in one or two days, which is long enough to arrive to adjacent islands. Compared with *E. multifasciata*, *A. sagrei* is more unlikely to disperse to other islands naturally because of the low saltwater tolerance and the currently limited distribution in Taiwan. We suggest focusing on human activities to prevent the further invasion of this introduced species.

## Supporting information

**S1 File. Current distribution of six species of lizards in Taiwan and the historical land connected with Asia Continent.** The Taiwan and adjacent islands with the coastline (A). The long dashed line represents the coastline during the Last Glacial Maximum 26.5–18 ka, and the short dashed line represents the coastline before 1.55 Ma. The green patterns represent the distribution of *P. elegans* (B), *E. longicaudata* (C), *D. swinhonis* (D), *H. frenatus* (E), *A. sagrei* (F) and *E. multifasciata* (G). The distribution of the species was referenced from The Atlas of Amphibians and Reptiles of Taiwan [68] and the Global Biodiversity Information Facility (http://www.gbif.org/). The coastlines were based on the open data from Natural Earth, which were licensed under the Public Domain.
(ZIP)

## Acknowledgments

We are grateful to Mr. CH Chang for the sample collection in the wild. We are also grateful to all members of WSH's lab for their help in both the laboratory and field work.

## Author Contributions

**Data curation:** Min-Hao Hsu, Jhan-Wei Lin, Jung-Ya Hsu.

**Formal analysis:** Chen-Pan Liao.

**Funding acquisition:** Wen-San Huang.

**Investigation:** Jhan-Wei Lin.

**Methodology:** Jung-Ya Hsu.

**Supervision:** Wen-San Huang.

**Writing – original draft:** Min-Hao Hsu, Wen-San Huang.

**Writing – review & editing:** Chen-Pan Liao, Wen-San Huang.

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
