## [Decision Letter · Decision Letter 0]

20 Jan 2021

PONE-D-20-33708

Over-ocean dispersal inferred from the saltwater tolerance of lizards from Taiwan

PLOS ONE

Dear Dr. Huang,

Thank you for submitting your manuscript to PLOS ONE. After careful consideration, we feel that it has merit but does not fully meet PLOS ONE’s publication criteria as it currently stands. Therefore, we invite you to submit a revised version of the manuscript that addresses the points raised during the review process.

We look forward to receiving your revised manuscript.

Kind regards,

Tzen-Yuh Chiang

Academic Editor

PLOS ONE

Journal Requirements:

3.We note that Figure(s) 4 and S1 in your submission contain map images which may be copyrighted. All PLOS content is published under the Creative Commons Attribution License (CC BY 4.0), which means that the manuscript, images, and Supporting Information files will be freely available online, and any third party is permitted to access, download, copy, distribute, and use these materials in any way, even commercially, with proper attribution. For these reasons, we cannot publish previously copyrighted maps or satellite images created using proprietary data, such as Google software (Google Maps, Street View, and Earth). For more information, see our copyright guidelines: http://journals.plos.org/plosone/s/licenses-and-copyright.

a.    You may seek permission from the original copyright holder of Figure(s) 4 and S1 to publish the content specifically under the CC BY 4.0 license. 

Reviewers' comments:

Reviewer's Responses to Questions

**Comments to the Author**

1. Is the manuscript technically sound, and do the data support the conclusions?

Reviewer #1: Yes

2. Has the statistical analysis been performed appropriately and rigorously? 

Reviewer #1: Yes

3. Have the authors made all data underlying the findings in their manuscript fully available?

Reviewer #1: Yes

4. Is the manuscript presented in an intelligible fashion and written in standard English?

Reviewer #1: Yes

5. Review Comments to the Author

Reviewer #1: The ms on dispersal inferred from the saltwater tolerance of lizards from Taiwan, is an interesting work worth publishing in PLOS ONE after revision of the points mentioned below.

Line 1: consider “trans-marine” instead of “over-ocean”

Line 50: “many introduced” change to “many species now recorded as introduced” or something similar

Line 52: delete “artificial”

Line 92: raft or artificial boats > rafts or boats

Line 92: Consider changing “In” with “Regarding” or similar

Line 106-107: I suggest rephrasing to “Taiwan and the adjacent islands are separated from the Asian continent by a sea strait with a depth of approximately 70 metres”. Additionally, “metres” is UK English style. If the journal requires US English you should change to “meters”.

Lines 135-143: Table 1. Please add a symbol (e.g. an asterisk) next to the name of the introduced species to facilitate readers not familiar with the local fauna. Adjust the legend accordingly.

Line 158: failed to lay eggs

Line 164: any individual observed having (oral?) secretions

Line 172: times

Line 179: in > on

Line 180: that individual directly contact with seawater. Please rephrase

Line 185-186: “The water surface slightly exceeded the abdomen of the lizard but did not exceed the

mouth”. How did you manage this? Did you have small rafts, or by controlling the water level in the tank to have it shallow enough for the animal’s feet to touch the bottom? Please clarify.

Line 192: Are there published data on the selected temperatures Tset (Hertz et al 1993)? It would be interesting as if e.g. species A has a preferred T=36oC it would maybe react differently than species B with a preferred temperature of 31oC

Line 196: Either 6 or 7.

Line 199: delete “in this study”

Lines 199-200: the incubation > their respective incubation

Line 208: delete “quite”

Lines 233-235: Notably…treatment. The phrase as is, is difficult to understand. Please rephrase

Lines 269-271: Not sure this is a correct approach. Most individuals of at least 2 species did survive. Why not include at least them in the comparisons? Moreover, even the weights of the dead specimen could be reported (maybe separately). Both info mentioned above would help understand if it was desiccation that caused deaths (i.e. severe weight loss) or another reason (e.g. possible intake of water orally causing salt water poisoning or other). You could at least add one or two lines commenting an this.

Line 336: …incubation rate. Do you mean hatching success?

Line 339: …hatchling success. Do you mean hatching success?

Line 346: arrive > arrived at

Line 348: … then connect this. Consider changing to “and possible connections of their physiological traits to their dispersal abilities” or something similar

Line 354: …if it could drift to Green Island and Orchid Island. I think this has no meaning, at least I do not understand it. Better omit or otherwise explain.

Lines 365-366: …that they – native. The phrase as is, is difficult to understand. Please rephrase

Line 341: Discussion. The entire discussion could be split in two sub paragraphs. The first till line 400 where you discuss the actual experiment’s results and the rest i.e. from line 401 onward where you discuss biogeographic scenarios.

Line 379: …than in the other species. Which other? the ones in the study? If yes, please specify

Line 418: drifting > oceanic drifts (actually, both “drifting” and “rafting” show energetic activity, whereas both are passive/involuntary. You could as well change “rafting” to “natural rafts”)

Line 423: distributes widely > have a wide distribution

Line 423: ocean > oceanic

Line 428: shortens the drifting > shorten the dispersal (or similar)

Line 449: is accordance > is in accordance

Line 451: disperses > disperse

Lines 451 - 453: Yes, but this does not exclude possible translocation by humans

Line 459: consider changing could to should

Line 463: delete “been”

Line 466: insert “the” before Kuroshio

References

Hertz P.E., Huey R.B., Stevenson R.D., 1993. Evaluating Temperature Regulation by Field-Active Ectotherms: The Fallacy of the Inappropriate Question. The American Naturalist, 142(5): 796-818)

6. PLOS authors have the option to publish the peer review history of their article (what does this mean?). If published, this will include your full peer review and any attached files.

Reviewer #1: No

---

## [Author Response · Author response to Decision Letter 0]

28 Jan 2021

Response to Reviewers

Response: We responded the additional requirements of the editor, the questions and the suggestions of the reviewer under each item.

Journal Requirements:

Response: We checked the manuscripts and files to meet PLOS ONE's style requirements.

Response: We had added descriptions about the permits for collection, husbandry and treatment procedures (Line 175-178) and uploaded PLOS ONE human endpoints checklist in previous resubmission. The full name of the authority that approved the field site access was checked again in this version.

3. We note that Figure(s) 4 and S1 in your submission contain map images which may be copyrighted. All PLOS content is published under the Creative Commons Attribution License (CC BY 4.0), which means that the manuscript, images, and Supporting Information files will be freely available online, and any third party is permitted to access, download, copy, distribute, and use these materials in any way, even commercially, with proper attribution. For these reasons, we cannot publish previously copyrighted maps or satellite images created using proprietary data, such as Google software (Google Maps, Street View, and Earth). For more information, see our copyright guidelines: http://journals.plos.org/plosone/s/licenses-and-copyright.

Response: In both Figure 4 and Supplementary S1 Fig, the coastlines were based on the open data from Natural Earth and Taiwan’s Ministry of the Interior, which were licensed under the Public Domain and the Open Government Data License (version 1.0; https://data.gov.tw/en/license), respectively. 

Response: We added the caption of Supporting Information files at the end of the manuscript as the requirement.

5. Review Comments to the Author

Response: We responded the questions and the suggestions of the reviewer under each item. We also added the annotation that included the suggestions of the reviewer and our responses. 

Reviewer #1: The ms on dispersal inferred from the saltwater tolerance of lizards from Taiwan, is an interesting work worth publishing in PLOS ONE after revision of the points mentioned below.

Line 1: consider “trans-marine” instead of “over-ocean”

Response: Revised as suggested.

Line 50: “many introduced” change to “many species now recorded as introduced” or something similar

Response: Revised as suggested.

Line 52: delete “artificial”

Response: Revised as suggested.

Line 92: raft or artificial boats > rafts or boats

Response: Revised as suggested.

Line 92: Consider changing “In” with “Regarding” or similar

Response: Revised as suggested.

Line 106-107: I suggest rephrasing to “Taiwan and the adjacent islands are separated from the Asian continent by a sea strait with a depth of approximately 70 metres”. Additionally, “metres” is UK English style. If the journal requires US English you should change to “meters”.

Response: We rephrased the sentence and changed the words of the UK English to US English as suggested. 

Lines 135-143: Table 1. Please add a symbol (e.g. an asterisk) next to the name of the introduced species to facilitate readers not familiar with the local fauna. Adjust the legend accordingly.

Response: We added the asterisks before the name of the introduced species in Table 1, and added the description at the end of the legend.

Line 158: failed to lay eggs

Response: Revised as suggested.

Line 164: any individual observed having (oral?) secretions

Response: Revised as suggested.

Line 172: times

Response: Revised as suggested.

Line 179: in > on

Response: Revised as suggested.

Line 180: that individual directly contact with seawater. Please rephrase

Response: We rephrased the sentence as suggested.

Line 185-186: “The water surface slightly exceeded the abdomen of the lizard but did not exceed the mouth”. How did you manage this? Did you have small rafts, or by controlling the water level in the tank to have it shallow enough for the animal’s feet to touch the bottom? Please clarify.

Response: We added the description as suggested.

Line 192: Are there published data on the selected temperatures Tset (Hertz et al 1993)? It would be interesting as if e.g. species A has a preferred T=36oC it would maybe react differently than species B with a preferred temperature of 31oC

Response: We choose this temperature because we found those species active well from 25-29 C in the wild. That’s why we use this specific temperature. We indeed do not have any data which species prefer in which temperature.

Line 196: Either 6 or 7.

Response: Revised as suggested.

Line 199: delete “in this study”

Response: Revised as suggested.

Lines 199-200: the incubation > their respective incubation

Response: Revised as suggested.

Line 208: delete “quite”

Response: Revised as suggested.

Lines 233-235: Notably…treatment. The phrase as is, is difficult to understand. Please rephrase

Response: We revised to make it easier to understand.

Lines 269-271: Not sure this is a correct approach. Most individuals of at least 2 species did survive. Why not include at least them in the comparisons? Moreover, even the weights of the dead specimen could be reported (maybe separately). Both info mentioned above would help understand if it was desiccation that caused deaths (i.e. severe weight loss) or another reason (e.g. possible intake of water orally causing salt water poisoning or other). You could at least add one or two lines commenting an this.

Response: Our experimental design was a factorial design, which lizard species, sex and salinity were three crossing factors. Because our results showed that the survival rates of four lizrad species were less than 30 %, to estimate those RTWLs were difficult, and it is impossible to include all SW results in the model duo to some completely empty cells. This is the reason we completely excluded the data of SW treatment in the model. However, we appended the RTWLs of the two survived species (E. longicaudata and E. multifasciata) in the same paragraph.

Line 336: …incubation rate. Do you mean hatching success?

Response: Revised as suggested. 

Line 339: …hatchling success. Do you mean hatching success?

Response: Revised as suggested.

Line 346: arrive > arrived at

Response: Revised as suggested.

Line 348: … then connect this. Consider changing to “and possible connections of their physiological traits to their dispersal abilities” or something similar

Response: Revised as suggested.

Line 354: …if it could drift to Green Island and Orchid Island. I think this has no meaning, at least I do not understand it. Better omit or otherwise explain.

Response: We deleted the description to avoid the possibility of misunderstanding.

Lines 365-366: …that they – native. The phrase as is, is difficult to understand. Please rephrase

Response: Revised as suggested.

Line 341: Discussion. The entire discussion could be split in two sub paragraphs. The first till line 400 where you discuss the actual experiment’s results and the rest i.e. from line 401 onward where you discuss biogeographic scenarios.

Response: We splitted discussion into two subsection by inserting two subsection header lines.

Line 379: …than in the other species. Which other? the ones in the study? If yes, please specify

Response: We added the description in the sentence.

Line 418: drifting > oceanic drifts (actually, both “drifting” and “rafting” show energetic activity, whereas both are passive/involuntary. You could as well change “rafting” to “natural rafts”)

Response: Revised as suggested.

Line 423: distributes widely > have a wide distribution

Response: Revised as suggested.

Line 423: ocean > oceanic

Response: Revised as suggested.

Line 428: shortens the drifting > shorten the dispersal (or similar)

Response: Revised as suggested.

Line 449: is accordance > is in accordance

Response: Revised as suggested.

Line 451: disperses > disperse

Response: Revised as suggested.

Lines 451 - 453: Yes, but this does not exclude possible translocation by humans

Response: We added the description after the sentence as suggested.

Line 459: consider changing could to should

Response: Revised as suggested.

Line 463: delete “been”

Response: Revised as suggested.

Line 466: insert “the” before Kuroshio

Response: Revised as suggested.

---

## [Editor Report · Decision Letter 1]

1 Feb 2021

Trans-marine dispersal inferred from the saltwater tolerance of lizards from Taiwan

PONE-D-20-33708R1

Dear Dr. Huang,

We’re pleased to inform you that your manuscript has been judged scientifically suitable for publication and will be formally accepted for publication once it meets all outstanding technical requirements.

Kind regards,

Tzen-Yuh Chiang

Academic Editor

PLOS ONE
---

## [Editor Report · Acceptance letter]

3 Feb 2021

PONE-D-20-33708R1 

Trans-marine dispersal inferred from the saltwater tolerance of lizards from Taiwan 

Dear Dr. Huang:

I'm pleased to inform you that your manuscript has been deemed suitable for publication in PLOS ONE. Congratulations! Your manuscript is now with our production department. 

Kind regards, 

on behalf of

Dr. Tzen-Yuh Chiang 

Academic Editor

PLOS ONE